

# The soil knowledge library (KLIB) - a structured literature database on soil process research

Hans-Jörg Vogel[1], Bibiana Betancur-Corredor[2], Leonard Franke[1], Sara König[1], Birgit Lang[2], Maik Lucas[1], Eva Rabot[4], Bastian Stößel[1], Ulrich Weller[1], Martin Wiesmeier[3], and Ute Wollschläger[1]

[1]Department of Soil System Science, Helmholtz Centre for Environmental Research - UFZ, Theodor-Lieser Str. 4, 06120 Halle (Saale), Germany
[2]Senckenberg Museum of Natural History Görlitz, Am Museum 1, 02826 Görlitz, Germany
[3]TUM School of Life Sciences Weihenstephan, Technical University of Munich, Emil-Ramann-Straße 2, 85354 Freising, Germany
[4] INRAE, Info&Sols, 45075, Orléans, France

Correspondence: Hans-Jörg Vogel (hans-joerg.vogel@ufz.de)

## 1 Abstract

In this technical note, we introduce a web-based application, the soil knowledge library (KLIB), for the compilation and classification of scientific publications on soil process research according to the specific site conditions and experimental boundary conditions. The tool was developed based on the understanding that experimental findings in soil science are highly dependent on soil type, land use and climate. The KLIB, therefore, goes beyond other available digital libraries by providing meta-data on the site conditions and experimental settings for each publication. A number of visualization tools have been developed to explore the contents of the literature database very efficiently in order to support and facilitate the literature search efforts of the users. The KLIB is designed as a collaborative effort to encourage soil scientists to participate by entering their own studies and extending the existing database.

## 2 Introduction

Soil is a highly complex system where a multitude of physical, chemical, and biological processes interact to generate a number of essential soil functions such as fertility for plant growth, water storage and water purification, nutrient cycling, carbon storage, and habitat for biodiversity. Today, we are still far from a solid mechanistic understanding of all these interacting processes. However, as a fundamental principle of science, identifying characteristic patterns of observed phenomena is a highly valuable initial step towards an in-depth process-based understanding of how soil as a complex system is functioning.

One example is the repeated observation that rather old and stable organic carbon in soil is frequently of microbiological origin (Miltner et al., 2012). This led to the insight that stabilizing carbon in soil is not mainly a result of chemical recalcitrance as previously thought but rather the consequence of physical protection within the heterogeneous soil structure as was already suspected by Schmidt et al. (2011).



During the last two decades, scientific tools were developed with the intention of providing more solid grounds for the observations of empirical relations. One approach is a systematic review of the available literature including a quantitative synthesis of research results. This approach, termed meta-analysis, was initially developed in medical research to synthesize data from multiple clinical trials (Koricheva et al., 2013). However, there are a number of critical issues coming along with meta-analysis. For example, the selection bias resulting from a purposeful selection of articles to synthesize and the publication bias reflecting the propensity for journals to publish studies with positive, hypothesis-affirming results (Haddaway et al., 2015). This also applies to meta-analyses in soil science, but there is an additional aspect that is especially critical in this field. In the medical realm, the object of interest, the human organism, is operating fundamentally the same way. In soil science, however, different soil types under different land uses and climates are "organisms" that have quite different internal controls and process interactions. Also, the temporal and spatial scale of experiments can have a substantial impact on experimental results. Hence, the integration of studies carried out at different locations under different boundary conditions might lead to inconclusive or even misleading results.

This problem of "comparing apples and oranges" might be the reason why different studies on the same subject come up with contradictory findings. For example, the impact of reduced tillage on the carbon level of soils was found to have a positive or negative effect, or no effect at all (Balesdent et al., 2000; Simpson et al., 2023). The impact of earthworm activity on the mineralization of organic matter can be accelerating (Wachendorf et al., 2014) or it may enhance carbon stabilization (Guggenberger et al., 1996). In this case, the difference between the studies is in the considered time scale. Earthworms foster microbial activity in their guts and within the fresh casts leading to increased mineralization rates. By contrast, in the long run, the organic matter within the relatively dense casts is stabilized with time since the accessibility for microbes is reduced. As another example Jarvis et al. (2013) demonstrated how the saturated hydraulic conductivity depends on site conditions and especially land use and cannot be considered as a function of soil texture. As obvious from these examples, studies that are considered in a joint meta-analysis need to be carefully pooled according to site characteristics and experimental boundary conditions.

Identifying characteristic patterns of observed phenomena is certainly a highly valuable initial step toward an in-depth process-based understanding of how soil as a complex system is functioning. It is also required for process-based, predictive modeling of how changing boundary conditions (e.g., in terms of land use or climate) will impact the multitude of soil functions. Such a model would be a major advance in soil science. Currently, we mostly rely on empirical correlations while our understanding of the relevant processes and interactions providing causality for the observed correlations is still limited.

In today's digital databases of scientific literature, which are accessible through various search engines, it is not possible to stratify the literature search according to soil types, site characteristics, or experimental boundary conditions. In addition, the search for specific processes and process interactions is difficult. The basic idea of the *Knowledge library* (KLIB) presented in this article is to allow for an analysis of the included research publications which can be structured according to site conditions, the investigated processes and the main results of the studies. It is mainly focused on soil process research but also methodological publications might be included. For process-oriented publications, all the meta-data need to be captured for





each individual study which is a considerable effort. For this reason, KLIB is organized as a community effort where many interested scientists can contribute with the idea that the work is shared.

In the following, we introduce the KLIB as an open-access tool for a structured analysis of published work on soil processes. We present examples of how to use KLIB in preparation for meta-analysis, and we discuss the potential of the KLIB to analyze published findings in general and how it might be helpful to identify research gaps.

## 3  The Knowledge-Library (KLIB)

### 3.1  Intention and functionality

The KLIB is a web-based tool accessible by any web browser (klibrary.bonares.de). It provides an interface for uploading relevant metadata related to scientific publications with a focus on soil process research. The main intention to develop this tool was the insight that observed soil behavior and the derived process understanding is highly sensitive to the boundary
conditions under which the individual studies were conducted. There are mainly two types of such boundary conditions: first, the specific site characteristics in terms of soil type, soil texture, land use, and climatic region, and second, the way how observations and measurements were made, more specifically, at which time and spatial scales, in the field or in the lab, using disturbed or undisturbed samples. The KLIB is a tool to record all this information for individual publications.

An additional but important category of information to be entered into the KLIB is the research question being investigated
in the publication, the metrics or measured soil characteristics used to investigate that question, and the key results that have been obtained from it. Entering all this information involves a not-inconsiderable amount of work. However, assuming that a publication wants to be read and understood anyway, the additional effort is not too much. Moreover, this needs to be done only once for a given publication and the result can be used by the entire scientific community and everyone can contribute, so, ideally, the work is shared among many colleagues.

The KLIB has a clear focus on publications dealing with experimental results that are suitable to improve our understanding of soil processes. Typical examples are: How do different types of tillage practices affect the physical, biological, or chemical properties of soils, and what site factors are important in this context? Or, how does a changing climate in terms of temperature and precipitation pattern impact the carbon balance in soils?

As the number of registered publications increases with time, the KLIB becomes an increasingly valuable tool for the
structured analysis of published results. To this end, an essential additional functionality of the KLIB is the provision of data filters and visualization tools for a targeted evaluation of the database. As an example, all publications that are dealing with the impact of zero-tillage on physical, chemical, and biological soil properties can be visualized in an interactive node graph panel. Because of the large number of publications, this graph can look pretty overloaded. Hence the various visualization tools can be filtered by, e.g., focusing on the comparison of conventional and zero-tillage and specific soil attributes such as soil carbon
or bulk density. In this way, the contents of the library can be efficiently explored. With the described functionality the KLIB





might improve the efficiency with which the scientific community can benefit from the continuously increasing number of publications.

## 3.2 Input of metadata

The bibliographic data of each publication including the abstracts are automatically extracted from the PDF file of a publication
by just dropping the icon of the file into the KLIB web application. In some cases where this is not possible, e.g., if the PDF document of a publication was created by scanning a paper document into a bit-map format, these data can be entered manually or extracted from a corresponding BibTeX file. For legal reasons, the original PDF file is not stored and made available in the KLIB. Afterward, the required metadata on site characteristics, sampling design, and key findings can be entered with the help of an intuitive user interface (Fig. 1). For experimental research articles, which are the main focus of the KLIB, a number of
different categories of metadata can be provided (if available in the publication):

a) **Type of study:** Is it an experimental study, a model application, a methodological study, or a meta-analysis? And in the case of an experiment: Does the study refer to individual sites or multiple sites? Is it a field or a lab experiment? What is the considered time scale?

b) **Soil & Site:** What are the soil types (with different classification schemes for selection: WRB, USDA, FAO, Ger-
man) and their texture? Which depths were investigated? Additional soil properties such as pH, organic carbon content, and bulk density can be provided optionally. What is the geographic region or spatial coordinate? What are climate characteristics (mean temperature, precipitation - could also be derived from the specified location)? What is the land use?

c) **Investigated Drivers:** In case the effect of different treatments in terms of soil management or physico-chemical
boundary conditions is investigated, these treatments considered as drivers can be entered here.

d) **Measured Variables:** Which soil properties are measured and/or monitored?

e) **Key Findings:** Here, the relation between the drivers previously given with the monitored properties or among different properties can be specified. These relations are provided in terms of simple effect categories (positive, negative, none, type of non-linear relations).

Almost all entries can be selected via a structured tree of predefined and standardized terms following the *AGROVOC Thesaurus* (Subirats-Coll et al., 2022). They were internally stored in a hierarchical tree of keywords which allows an efficient exploration of the database.

## 3.3 Exploring the database

All process-oriented publications included in the library are supplemented by metadata of different categories: Investigated
soil properties, soil management, and soil and site conditions. Besides a classical search for keywords, authors, and date of



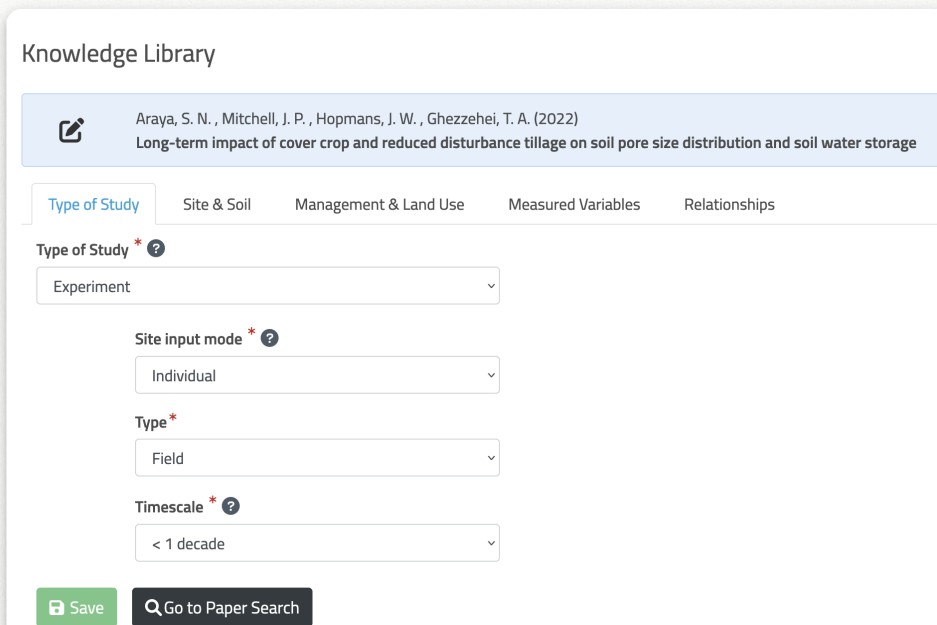

**Figure 1.** Web interface of KLIB to enter metadata on site characteristics, experimental design and key outcomes reported in a publication on soil process research. The required input is divided into the categories: Type of Study, Site & Soil, Management & Land use, Measured Variables and Key Findings.

publication, the KLIB offers the functionality to search its contents based on interactive graphical tools that are designed to visualize complex interrelations within and among the different categories. Five different visualization tools are provided. They are described in the following section, together with typical application examples.

*Keyword network (KN)*

This visualization displays the network of publications that are dealing with a specified combination of soil properties or management practices. It is possible to combine an arbitrary number of keywords. A large number may produce an overloaded network which, however, may depict the main focal points of the publications. An example is illustrated in Fig. 2A for a complex keyword network including physical and biological attributes together with all different tillage options. The two visual focal points in this graph are "zero-tillage" and "conventional tillage" which are obviously the two most frequently

addressed topics in the current content of KLIB. A simple example is shown in Fig. 2B and C where only the minimum of two attributes was selected, namely bulk density and conservation tillage. The latter is a "parent" category in the keyword tree containing the four "child nodes" ridge tillage, minimum tillage, zero tillage, and strip tillage which are visualized as well. In this case, there were no publications in the database dealing with bulk density and strip tillage, hence, this point is missing in the visualized network. The publication metadata can be accessed by a click on the grey circles connecting pairwise attributes.

The size of the circles corresponds to the number of publications that are provided as a list (Fig. 2D).



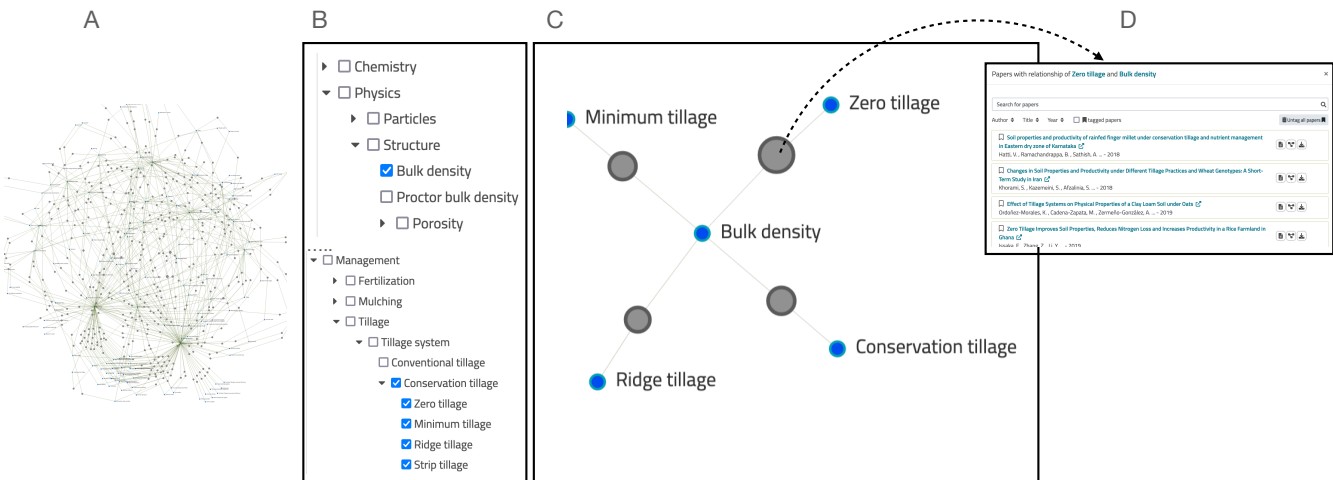

**Figure 2.** Example of a complex "Keyword network" including physical and biological attributes together with all different tillage options (A, details not relevant here) and a simple network (C) where only "bulk density" and "conservation tillage" was selected (B). When clicking on the grey circles the list of publications dealing with the connected attributes is shown (D).

*Properties affected by management (PM)*

This visualization mode is focused on a single soil property such as soil organic carbon, bulk density, or earthworm abundance. It displays management practices that affect this specific property. The visualized network allows the identification of publications that compare a set of different management practices with respect to the selected property. An example of bulk

density is presented in Fig. 3A showing that the impact of conventional tillage and zero tillage on bulk density are often compared to other tillage systems. Another example of earthworm abundance affected by different tillage practices is shown in Fig. 3B. By default, all relevant management practices are plotted but this can be narrowed down to a convenient subset by the user. Again, the list of individual publications can be found with a click on the grey circles, where the key findings of each individual publication with respect to the chosen soil property can be found as well.

*Drivers affecting properties (DP)* This visualization mode is complementary to the PM mode. It focuses on drivers which can be either management practices, soil properties or site conditions, and displays which soil properties are affected by a chosen driver. In addition, the publications that deal with other drivers that were found to impact the same properties are considered as well in the illustrated network. This can lead to rather complex networks as shown in Fig. 4A for "conventional tillage" as the chosen driver. There is a large number of soil properties affected by conventional tillage, as well as a large number of other

management practices that were investigated in comparison to conventional tillage. If this is reduced to only "zero tillage", the structure of the graph becomes much less complex (Fig. 4B). In this way, references for the impact of different drivers can be identified to compare their results. Another simple example is shown in Fig. 4C for the properties affected by organic fertilizers




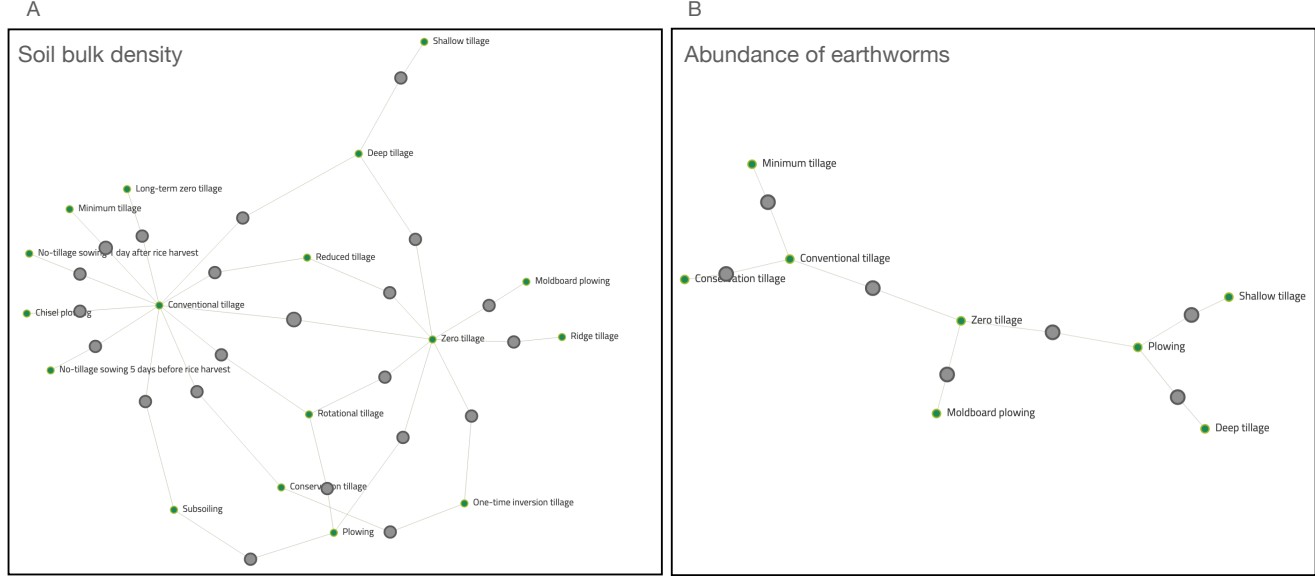

**Figure 3.** Visualization mode "Properties affected by management" (PM) for bulk density (A) and earthworm abundance (B) showing which management options are compared with respect to their impact on the chosen soil property. When clicking on the grey circles the corresponding list of publications together with their key findings is shown.

in comparison to inorganic ones. Of course, also this visualization might become more complex as the number of publications included in the KLIB increases.

*Property-property relation (PP)*  This visualization mode provides relationships between measured soil properties. Such comparisons typically look for correlations between soil properties rather than causal relationships between a particular driver and selected soil properties as in the previous visualization tools. An example is illustrated in Fig. 5A where soil organic carbon was chosen by the user and all other properties that have been investigated in combination with it are illustrated in the graph. The related publications are shown via grey circles. The categories of displayed properties can be reduced according to the

interests of the user.

*Publication-centered visualization (PC)* This visualization mode can be accessed for each individual publication. It illustrates all soil properties and drivers addressed in the given publication which is placed in the center of the graph (Fig. 5B). Other publications that share at least two of these properties and drivers are shown as well. The related publications are shown via the grey circles linked to those aspects. This visualization tool is designed to search for other publications dealing with closely

related topics.





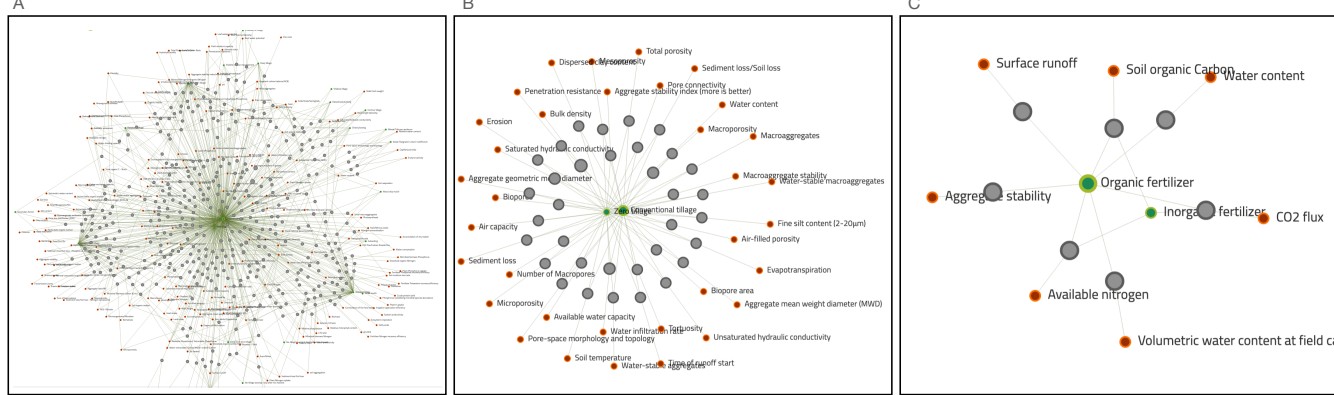

**Figure 4.** Visualization mode "Drivers affecting properties" (DP). Plot A is the result for the selected driver "conventional tillage" (in the center) when all other drivers which were investigated with respect to the same properties as conventional tillage are plotted. In plot B this graph is reduced to conventional tillage vs zero tillage only. Plot C illustrates the soil properties affected by organic fertilizers vs inorganic fertilizers. When clicking on the grey circles the corresponding list of publications together with their key findings is shown.

For any visualization mode, the results can be filtered with respect to soil and site conditions. It is possible to reduce the visualized results to specific soil properties like texture classes, pH ranges, or carbon contents, or to site properties like climate zones.

Each visualization tool leads to a selection of items that match the criteria being searched for. In all visualization tools, this
selection is reached via the grey circles in the graphics and is presented as a list of publications (Fig. 6). For each publication on this list, the key findings are provided through intuitive symbols (e.g. "**+**", "**-**", "**x**", for positive, negative, or no relation, respectively). This corresponds to what was originally indicated as "key findings" when the publication was entered into the KLIB. Moreover, additional information on each publication is provided. The title of the publication is linked to the website of the publisher and the symbol directly after the title provides a link to *Google Scholar* for this paper. By clicking on various
intuitive symbols associated with each publication the user can access all information stored for each publication. Alternatively, the publication-centered visualization mode can be invoked. Additional options are available to the user who originally entered the item into the library. The user can edit the information about this publication, remove the publication from the library or share the rights to edit the content with other users. The latter can be very helpful when working in collaborative groups. An example is illustrated in (Fig. 6).
All search results can be downloaded in the open standard file format *json* including all information stored for each individual publication.

## 3.4   Potential applications

The KLIB can be used in various ways, which are briefly outlined below.




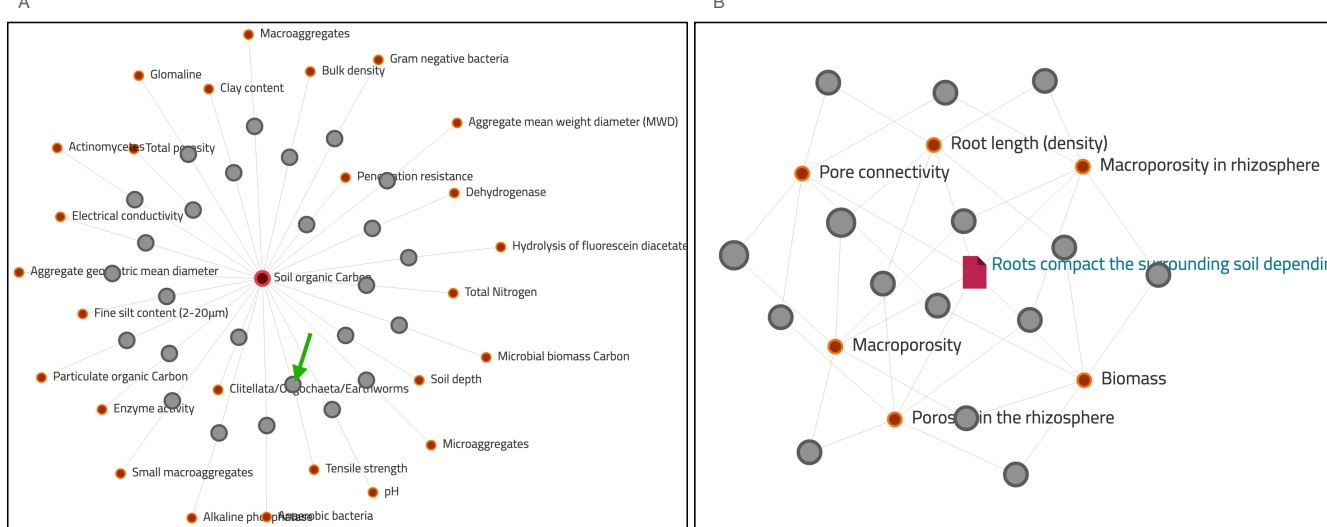

**Figure 5.** Visualization mode "Property-property relation" (PP) displays all soil properties which were investigated together with some chosen property. The left graph (plot A) shows an example of soil organic matter as chosen property. The "publication-centered visualization" (plot B) displays all soil properties (orange circles) which are investigated in a chosen publication (red rectangle in the center) and provide all other publications that share at least two of the investigated properties of the chosen publication. In any case, the identified publications can be accessed through the grey circles connecting those properties. As an example, the publication list generated after choosing the grey circle marked by the green arrow is shown in Fig. 6

### Basic literature search

A basic mode of using the KLIB is for the search for publications dealing with a specific subject. Thereby, the search is not only reduced to keywords and their possible combinations. The search can be highly targeted to specific interactions between soil properties or to the effect of different drivers on soil properties. A special feature is the direct access to the metadata on site conditions for each publication. This allows the search for literature to be specified for selected site conditions, such as soil type, climate zone or vegetation.

A good example of a current research question is to search for published findings on the influence of reduced tillage on stabilized soil organic carbon. It is expected that some studies show a clear positive effect compared to conventional tillage, but also others have found no or even a negative effect. The difference may lie in the site conditions or the specific experimental settings, such as soil type, initial carbon content, the time scale of the experiments, crop rotations studied and climatic conditions. The KLIB provides the means to structure published findings according to relevant boundary conditions and to find new

related literature.

### Conducting systematic reviews and meta-analysis





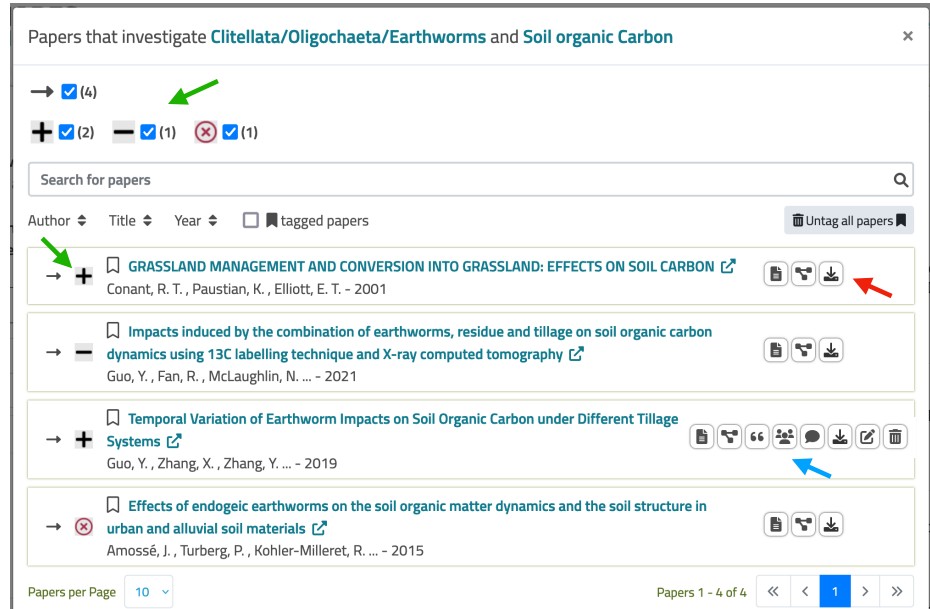

**Figure 6.** Typical example for a list of publications dealing with the relationship between earthworm abundances and soil organic matter (Fig. 5A). The type of relationship is indicated for each individual publication and summarized for all publications on top (green arrows). The symbols behind each publication (red arrow) provide access to i) the abstract and a summary of all metadata of the publication, ii) the publication-centered visualization, and iii) the download of all information related to this publication. For the user who has entered the publication into the library, there are additional options (blue arrow): i) share the rights to edit with other users, ii) add and edit comments to the publication, edit the publication information or iii) remove the publication from the library.

The main objective of hypothesis-driven meta-analyses is to generate a quantitative estimate of a studied phenomenon, i.e., the effectiveness of an intervention, by gathering data from independent primary studies that focus on the same research question. This can only be accomplished by evidence consolidation from multiple studies that allow for side-by-side comparison and by filtering out the signal from the noise (Mikolajewicz and Komarova, 2019). For this purpose, the KLIB provides the necessary tools to stratify the search results along critical boundary conditions. An example would be a meta-analysis on the influence of earthworms on the saturated hydraulic conductivity of the soil. Based on a number of publications, a meta-analysis can be conducted to estimate the correlation of earthworm abundance with measured hydraulic conductivity. Considering that sandy soils with larger particle sizes typically have a much higher hydraulic conductivity than fine-textured soils, the results of the meta-analysis will likely lead to contradictory results if the results presented in the included publications are not analyzed separately for different soil textures. This is because at the same time, sandy soils are often an unfavorable habitat for earthworms because of their reduced water-holding capacity and the reduced mechanical stability of the soil structure. Hence, a meta-analysis would be potentially biased by the results presented in studies conducted in sandy soils, which will likely report





increased hydraulic conductivity and reduced earthworm abundance. In this case, the conclusions with respect to the original

research question could be completely wrong. The KLIB provides the tools to systematically detect potentially misleading results from the initial literature search and find alternative ways to avoid them by providing information on critical boundary conditions.

The procedure of meta-analysis starts with the formulation of a question or defining a scope. It can either be for the evaluation of a theoretical hypothesis (focus on causality) or a test for a consistent relationship between two or more variables (focus on

correlations). Hence, when conducting a meta-analysis, it is important not to mix studies that report observed relationships and relationships identified through experimental manipulation (Cote and Jennions, 2013). The KLIB can help to easily identify these different types of studies, as for each publication the type of relation between drivers and monitored properties is clearly specified by the user.

In most cases for soil science research, the key finding of a meta-analysis is the mean effect of a certain intervention, whether

it differs from the null expectation, and whether the variation of the outcome of the different studies can be explained. The question regarding the variation invites the meta-analyst to think about factors that differ among studies and can affect the estimated mean effect. At this stage, the KLIB can help the meta-analyst to categorize or search for studies among different important groupings like site conditions, climate, management-related factors, type of study (field or lab), duration of the study, methodology used to measure outcomes, among many other factors.

Once the research questions are formulated, a protocol (i.e., following the PRISMA framework (Page et al., 2021)) is needed for meta-analysis, which will specify an objective search strategy, identify the inclusion criteria for the studies and discuss potential sources of heterogeneity. Writing such a protocol is a crucial step as it is intended to reduce biases in data collection. The KLIB can not provide a complete and comprehensive list of references for meta-analysis but it can assist the meta-analyst with the elaboration of the protocol in several ways: (i) search and compilation of a list of potential publications

that could be included in the meta-analysis and will help with the identification of study inclusion criteria; (ii) identification of different methods used in the studies; (iii) identification of site-related sources of heterogeneity of the results; (iv) identification of studies with extractable data in terms of data quality, as the KLIB already presents available metadata on the publications; (v) serve as a filter for study quality, as the publications uploaded to the KLIB have already been screened by peers of the scientific community.

The KLIB can also assist the meta-analyst in more advanced stages of the meta-analysis, i.e., during the scoping search. The purpose of a scoping search is not to find every study on a particular topic, but to identify potential moderators, decide the criteria that would make a study relevant or irrelevant, identify suitable outcome measures and make exclusion decisions based on title, abstract, or site-related features. The KLIB makes the relevant metadata for publications easily accessible, which will facilitate the scoping search. Additionally, with the network visualization tools, clusters of related publications can be easily

identified which can also speed up the process of searching and decision-making. The KLIB has been used in the past to support the scoping searches, as well as for the identification of potential moderators and outcome measures in Betancur-Corredor et al. (2022a, b); Lang et al. (2023).



*Improving process understanding*

Another motivation to implement the KLIB was the idea to improve process understanding. Many experimental studies in soil
that delve into the impact of various drivers on soil properties or into the interaction between different soil properties lead to
correlative connections between the investigated quantities after an adequate statistical analysis of the observations. However,
the ultimate goal in most cases is to come up with a better understanding of causalities and their underlying processes. This
is particularly true if we are striving for mechanistic, process-based models that are capable of predicting the changes in soil
systems in response to external drivers of land use and climatic conditions.

By filtering the available literature according to critical experimental boundary conditions, the KLIB can help to reduce
the complexity of the reported results, opening the view to the underlying causal relationships. The example of the impact of
reduced tillage on soil carbon given above can also be viewed from this perspective, for instance by identifying one specific
driver - such as site conditions or management - to explain the major differences in contradictory study results. This joint driver
might give a hint about the underlying processes responsible for the dynamics of soil carbon under different tillage regimes,
and could then be further tested or integrated in the model description.

*Structuring references for individual projects*

All three different modes of application are basically useful for any scientific work, be it a classical scientific publication or
a Ph.D. thesis. It is an essential part of embedding one's own research into the broader research landscape. Hence, the KLIB
can be used for preparing individual publications, project proposals, or theses. It is possible to restrict all visible contents and
visualization tools to only those publications that were entered by a single user or a group of authors. In this way, the KLIB
can be used as a tool to structure the bibliography of a specific project.

## 3.5 Technical implementation

The KLIB is a web-based application that follows modern standards and techniques. For the frontend we are using the frame-
work *Angular* as a foundation for all continuing functions (ang). The backend is built on *node.js* with *Express.js* to handle all
the data structure and the connection to the database. While being visible and searchable for all users, all entered papers are
connected to a unique login of each person, which gives editing rights only to authors and those who have been given access.
The process of entering bibliographic data is supported by the option to extract the DOI from a PDF and retrieve the data from
the *Crossref-API*. This allows us to get all information in seconds without the need to enter it manually.

To provide safety, we are using newest standards and techniques like *jsonwebtokens* for authenticating to the KLIB and on
backend site *MongoDB* to store the data secured. The entire database is backed up several times a day, so that, in the event of
a problem, all data entered can be restored to the most recent version possible. *MongoDB* provides the speed to prepare the
data for complex visualizations in an acceptable amount of time. The *JSON* format is commonly used to store data and send it
through several API calls to the user on the frontend side.

We implemented complex algorithms to generate the information for the visualizations and build the different networks that
can be explored and adjusted. The presentation itself is handled by the *D3.js* library, which provides powerful tools to generate



smooth networks. The implementation of canvas as the rendering structure is essential for handling big datasets and many points in a network, the conventionally used *svg* is too slow for our application and causes performance problems on many devices.

The whole structure of our KLIB is built to be easily accessible and a framework like Angular keeps the whole structure modular and easy to update. Any new features or extra visualizations can be implemented at the need of the users or edited according to their feedback.

## 4  Discussion and outlook

With the KLIB, we offer a tool for structuring soil process research relying on relevant publications including metadata such as site characteristics and experimental boundary conditions. The KLIB can be used freely by all interested in soil science and its value and usefulness will grow with an increasing number of active users and uploaded publications. At the moment, about 600 publications are part of the KLIB. This is sufficient to assess the functionality but for many questions, this is a still far too small number to provide a good overview of the published knowledge. Therefore, all soil scientists are invited to contribute and ultimately benefit from KLIB. The visualization modes that make the most of the library are designed to be efficiently applied when the number of publications included in KLIB grows substantially, which is actually the intended objective.

At the moment the KLIB is designed for experimental work directed towards a better understanding of soil processes that depend on site characteristics and experimental boundary conditions. The development was motivated by the need for mechanistic soil models (Vogel et al., 2018), such as Expert-N (Engel and Priesack, 1993) or APSIM (Holzworth et al., 2014) that are designed to describe and predict changes in soil functionality based on a sound understanding of the underlying processes. However, these processes and their interactions can hardly be described based on first principles. Instead, their parameterization needs to rely on empirical findings documented in publications on soil process research. The KLIB should be a valuable tool to identify processes and to come up with ideas on how to represent these processes in systemic models. In fact, only a smaller proportion of publications within soil science are related to process research. A very substantial part deals with methodological developments. In the future, the KLIB will be further developed for such publications as well, in order to also support the literature search for methodological aspects.

Of course, an open-access tool like KLIB raises the question of how to ensure the quality of the data entered by many different users from different fields of soil science. One quality assurance measure is that only registered users can upload publications including metadata. It is completely transparent who has uploaded which publication to the library, which should support the responsible use of the KLIB. The KLIB is maintained by the BonaRes Centre to quickly remove obvious errors. Further evolution will show whether there is a need for additional functionality to allow all users to mark suspected inconsistent entries as such.

A critical obstacle to uploading publications is the considerable effort required to enter the metadata for each publication, although the user interface supports this process very well. Many entries can be made very quickly via drop-down menus,



taking into account different classification schemes and metrics with different physical units. As a side note, this effort is significantly reduced for well-written articles where the information can be found easily.

Despite all this, it seems somewhat anachronistic that in times of rapid development of artificial intelligence, we are still entering such metadata via the computer keyboard. We are currently testing advanced text-mining tools as done recently by Blanchy et al. (2023) to minimize the need for manual input of data in perspective. This can work quite well if the authors stick to existing classification systems and nomenclature. The practice has shown, however, that there is an enormous diversity in terminology and that it will take some development before self-learning systems can digest this diversity in such a way that the

results meet the requirements of KLIB.

Today, there is a number of web-based tools available which are useful to explore published work in a given research field. Tools such as "CONNECTED PAPERS" (https://www.connectedpapers.com) or "Citation Gecko" (https://citationgecko.azurewebsites.net/) start from a number of seed publications to analyze which other publications have been published on the same subject and how the various publications cite each other. Other tools such as "paper-digest" (www.paper-digest.com) analyze the contents of

individual publications using methods of text mining to summarize their contents. The underlying concepts are developing fast at the moment and the KLIB will certainly profit from this to automatically extract information on the spatial coordinates, the soil type, the land use and other boundary conditions in the future. To the best of our knowledge, the KLIB is the only tool that can be used to relate scientific publications and their major findings directly to site conditions, which is of crucial importance for their interpretation, especially in soil science. Another unique feature of the KLIB is that it can be used by individual users

for their own work, but the information they provide is also accessible to other users as well. In this way, we hope that the KLIB will develop into a community venture from which we can all benefit.

## 5  Acknowledgements

We would like to thank the many colleagues who gave us a lot of advice on the design of the user interface and functionality during the development of KLIB. This project is funded by the German Federal Ministry of Education and Research (BMBF)

in the framework of the funding measure "Soil as a Sustainable Resource for the Bioeconomy – BonaRes", project "BonaRes (Module B): BonaRes Centre for Soil Research".

*Author contributions.* Conceptualization: HJV, BS and UW . Data curation: HJV, BBC, SK, BL, ML, ER, BS and MW. Technical implementation: LF and UW. Writing original draft: HJV and BBC. Writing review and editing: all.

*Competing interests.* There are no competing interests



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
