# Peer review of "The soil knowledge library (KLIB) - a structured literature database on soil process research"

_EGUsphere, 2023_

## Author Comment (AC1)

*We would like to thank the reviewer for the constructive comments and suggestions. Below we explain how we will address these suggestions. Our responses are in italics.*

1. There are quite a lot of grammatical and style issues with the presentation that distract from ease of reading. There are also many examples of non-scientific terms such as 'pretty' that are going to confuse people for whom English is a second language.

*We will revise the language and choose less sloppy formulations*

2. The rationale for the tool is good, and is well explained.

*We are glad that we could explain the rational of the tool satisfactorily*

3. Paragraph beginning line 69: several statements are made that read more as opinions instead of evidence. Given that this is where you are justifying a large part of the effort required for involvement, I think your statements need to have good evidence here.

*We are not sure about this comment. In this part of the manuscript we describe how the KLIB is structured and what our intensions are to do so. These are not opinions, we just explain why we have structured the KLIB the way it is. We will qualify the few statements in this context to make clear that this is our interpretation of the situation (e.g. that the additional effort for the KLIB is not too big if the user has red and understood a paper, or that the KLIB will become more valuable the more paper it contains).*

4. There are probably as many different ways to design this kind of system as there are soil scientists. Please give a bit more justification for the structure of the overall system, particularly in terms of which factors are considered important for data entry.

*We will add a paragraph on the major motivation for developing the Klib which mainly comes from mechanistic soil modeling where many processes are not known in detail and we need to rely on empirical findings reported for well-defined experiments.*

5. The Technical Implementation section needs to come much earlier, so that the reader can visualise not just what you have done, but how you have done it. I suggest having it as section 3.2.

*We agree and will move the technical Implementation section to the position 3.2*

6. The Technical Implementation section does not contain enough detail for the work to be duplicated.

*We do not believe that anyone wants to duplicate the KLIB. After all, it is publicly available and can be used by anyone. Therefore, we limit the description to the used software tools, which might be important for those who want to program similar applications.*

---

## Author Comment (AC2)

*We would like to thank the reviewer for the constructive comments and suggestions. Below we explain how we will address these suggestions. Our responses are in italics.*

Thank you for letting me review this manuscript. I would like to congratulate the Authors on the good work put into this manuscript and overall on the KLIB project!

The article is well written and the structure is adapted to present a tool (the KLIB) that is certainly useful for the soil science community. I myself used this tool in several projects and I am happy to see it further developed! I agree with the authors that often the outcomes of a specific agricultural practice are bound to specific pedo-climatic conditions. To my knowledge, the KLIB is one of the few (if not the only one) tools that enable us to access this pedo-climatic information in a structured way. The authors of the KLIB propose, based on publications, to extract this information (manually for now, more automatically in the future) from papers and make it available in different visualization tools based on network graphs. This is certainly of interest at times where the scientific literature is growing quickly. I hope this publication will give it more visibility.

*Thank you for this very positive assessment - we definitely share your hope!*

L7: " a number of visualization tools": maybe be more specific with some examples?

*We will add examples*

Abstract: I would directly add the URL of the KLIB so interested person can have a look at it directly from the abstract

*Good point! We will do*

L47: 'correlation' is repeated twice in the sentence

*Will be corrected*

L71 double negation: "not-inconsiderable", maybe "not negligible" or "considerable"

*We change this to simply "considerable"*

L80: I agree, we need a way to report information with structure and the KLIB is a good option for that (among other things)

*Yes!*

L102: for organic carbon: is it possible to also specify the method used (and possibly the date of the sampling)? values can varies quite a lot between methods

*Yes, good point! For the next update we will add a free text input field where the method can be specified*

L105: I would give a few examples of a drivers and measured variables already here in parenthesis

*We will do!*

L110: lovely! I like standard terms!

*Yes!*

L110: does the tree enable intermediary terms selection? For instance if I know pH was measured but there is no information in the paper if it was with the pH KCl or pH H2O method, can I still add it as a more global level? like just pH? And in this case, if I select just 'pH' in the keyword tree, will I be able to see the publications for both 'pH KCl' and 'pH H2O'? I miss the information about how it works with the 'levels' of the keywords tree throughout the KLIB

*We will explain this in more detail: "The entries can be chosen from different hierarchical levels depending on the availability of information. For example, it is possible to choose "pH" or the following level either "pH(water)", "pH(KCl), "pH(CaCl$_2$). In visualization mode the search for "pH" will show all the possible entries at the following level.*

fig1 is not referenced in the text

*We will do*

fig2c I always find it strange that the property and drivers have the same color (of the dot). Wouldn't it make sense to separate drivers and measured variables? or is that intentional?

*We already changed this and now use different colors for properties and management options. The figure will be updated*

fig2 and fig1: add a date when the screenshot was taken (as it might evolved in the future I guess) .. for next figures too

*We will do*

PC mode and others mode: there is always a "share at least 2 properties" rule, can this be modified (to 3 or 4 properties for instance?) - this might be a future feature request

*We introduced this rule because the visualized network would be too crowded if we go down to just one shared property. But of course those that share 3 or 4 properties are shown as well and can be identified by the number of lines connected to each grey node. If we find in the future that we need to increase this filter to 3 or 4 shared papers we will do that.*

L175: why not also add a .csv output? (easier for people working in spreadsheet) to be able to download the entire list and not just entry per entry (a JSON/CSV combination can be nice)?

*Unfortunately, CSV output is not possible because the data is very complex and consists of many nested objects, it would not be readable either by human or a script*

L185: this is a good example. But it supposes that the user knows (or tries a few times) to see on which pedo-climatic variables the relationships vary. Isn't a way to show (a bit similar to a meta-analysis moderator figure but in a qualitative way) the direction of a relationship according to different pedo-climatic variables. We could imagine we select 1 driver (and a

reference possibly), 1 soil variable and then several pedo-climatic variables and we make a figure of all that?

*We thought about this problem but did not find a good solution yet. The required visualization would have at least 3 dimensions (measured properties, driver, pedo-climatic variables). At the moment you get a list of all papers that report about relation between some driver and a property. On top of this list it is indicated how many papers found a positive, negative or no relation and you can select the respective papers by one click. This allows a rapid qualitative overview on the pedoclimatic variables that might be relevant for a particular meta-analysis. Otherwise, each visualization can help the analyst to narrow down the search to some highly relevant and sufficiently reported pedo-climatic variables. But as you said, this requires some prior knowledge (or hypothesis) or the analyst may have to try some times. In short, we would be happy to receive any suggestions how this can be improved.*

L210: "it is important not to mix ... experimental manipulation": this sentence and how it articulates with the previous one is a bit confusing for me. Are the "relationships identified through experimental manipulation" linked with "theoretic hypothesis (focus on causality)", if so I suggest switching the order of one or the other sentence so that it is consistent between sentences

*Good point! We will switch the order of the second sentence and we will mention that experimental manipulations are typically searching for causal relations This paragraph will be then rephrased as follows: The process of conducting a meta-analysis begins by formulating a question or defining the scope. This can be done to evaluate a theoretical hypothesis, focusing on causality, or to test for a consistent relationship between two or more variables, focusing on correlations. Therefore, it is crucial to avoid mixing studies that report observed relationships with those that identify relationships through experimental manipulation (Cote and Jennions, 2013). The KLIB can assist in distinguishing between correlational and causal studies since the user specifies the type of relationship between drivers and monitored properties for each publication.*

L217: I agree that KLIB can help access this information but I still find it difficult to have an overview of the pedo-climatic factors and their influence on a given soil-property vs practice relationships. KLIB enable filtering and then we can explore each list of publication individually but we don't have an 'overview' (similar to a moderator figure you can find in meta-analysis when an effect size is decomposed per soil group for instance) see also L185

*Yes, we are aware of the usefulness of having such an overview (see comment above for L185). Unfortunately, at this stage of the KLIB development, it is not yet possible to have an overview (quantitative or otherwise) of the effects reported in each individual study. We aim to highlight in the manuscript the role of the KLIB as a tool for categorization/classification and search of studies to support the meta-analyst at early stages of the process. However, we trust that in later stages of development such visualizations will be possible, in the form of vote counting, for example (for positive or negative effects).*

L229: "screen by pairs of the scientific community" what is meant by this? only pair-review publications are accepted or is there another kind of filtering behind? according to which criteria? Is there a double quality control on new entries inserted?

*It simply means that the papers in the KLIB already passed a peer review process. We will replace "screened" by "reviewed"*

section 3.5: nice to have the implementation description. I was wondering why you went for a mongoDB (unstructured database) rather than a (traditional) relational one? Also what kind of consistency control does mongoDB implement? (relational have "ACID checks" -- I am not an expert on this)

*MongoDB is used for performance reasons and because it is very flexible to changes and optimizations of the structure. There are several data layers to ensure consistency of the data but also keeping it structured, in the end we are using a construct that is similar to relational databases in these terms while keeping the flexibility and other advantages of MongoDB.*

L295: I was wondering if it wouldn't also be interesting to encourage manuscript's author to add their publications to the KLIB? Another proposition could be to enable a "second check" by another user upon entering data in the KLIB to ensure quality control (but I understand it takes time). Working with the transparency of who added the entry is already a good start as you mentioned.

*It is our plan to approach soil related journals to provide a link to the KLIB once a paper is accepted – at this point in time, it should be very easy for the author to add his/her paper to the KLIB. We hope this will improve the contents and quality of the KLIB.*

L301: I would also add the fact that a cookbook page is also available on the website and serve as documentation for adding new entries

*We will do*

L331: Angular reference, a bit strange to cite it like this but why not. Then other frameworks should be cited too I guess no?

*The only other framework used and will be cited is bootstrap - https://getbootstrap.com/*

Open questions:

The database collected in this work is certainly of interest and can be used for further research. I think it can be good to add a way to download this database and list of selected papers. Being able to explore it through the different visualizations is already very nice but does not allow for more automated methods to be applied.

*It is already possible to tag selected papers. For examples those that were found for a specific search or a combination of various searches, or all papers you provided. Then on the main page you can select "tagged papers" and download this list in json format.*

Does the KLIB have a DOI itself? Or shall we cite this paper if we want to use it? I suggest adding a "how to cite" section to the website maybe in the 'info' tab. This can be added to a 'data availability' section of this manuscript.

*Good point! There is no DOI, the KLIB should be cited via this paper together with the weblink – we will add a "how to cite" section to the website.*

Is the KLIB only based on field experiments?

*It is focused on field and lab experiments looking at soil processes and interaction between soil properties. We will extend the part on methodological and modelling studies which is already available but only in a rudimentary mode.*

How about review papers or meta-analyses? Can they be entered in the database or is it mainly "primary studies" that are seeked?

*They can be entered and need to be indicated as such (this is the very frist entry on the first page "Type of Study"). Of course it is difficult how to deal with soil & site properties in case of meta-analysis but we will try to improve this in the future.*

If a publication contains two experiments made on different soil types (so two sites) and that different relationships have been found for each type, should it be entered twice so that the two different relationships can be added?

*You have very well found the critical points! It is possible to enter a number of different soils and sites if it is appropriate for the given paper. Then, in the results section you can select this spoils/site to provide the relationships individually. We did not explain this in detail in the paper and thought this will be a functionality for professional power users…*

Bug: sometimes the link to Google Scholar is not populated. How to reproduce: in 'Properties affected by management', type 'bulk density', then select one of the gray dots and click on the "external link icon" for Google Scholar.

*It was a bug and is now fixed*